# Repurposing Biomolecules from *Aerva javanica* Against DDX3X in LAML: A Computer-Aided Therapeutic Approach

**DOI:** 10.3390/ijms26125445

**Published:** 2025-06-06

**Authors:** Abdulaziz Asiri, Abdulwahed Alrehaily, Amer Al Ali, Mohammed H. Abu-Alghayth, Munazzah Tasleem

**Affiliations:** 1Department of Medical Laboratory Sciences, College of Applied Medical Sciences, University of Bisha, 255, Al Nakhil, Bisha 67714, Saudi Arabia; amfasiri@ub.edu.sa (A.A.); ameralali@ub.edu.sa (A.A.A.); mhahmad@ub.edu.sa (M.H.A.-A.); 2Biology Department, Faculty of Science, Islamic University of Madinah, Madinah 42351, Saudi Arabia; 3Department of Public Health, College of Applied Medical Sciences in Al-Namas, University of Bisha, Al-Namas City 67714, Saudi Arabia

**Keywords:** acute myeloid leukemia (LAML), DEAD-box helicase 3 X-linked (*DDX3X*), *A. javanica*, ADMET, molecular docking

## Abstract

Acute Myeloid Leukemia (LAML) is a life-threatening hematological malignancy, and the DEAD-box helicase 3 X-linked (*DDX3X*) gene is a potential yet underexplored target gene for LAML. Biomolecules derived from medicinal plants like *Aerva javanica* offer a great source of therapeutic candidates. This study aimed to investigate the role of *DDX3X* in LAML and identify plant-derived biomolecules that could inhibit *DDX3X* using computational approaches. Pan-cancer mutational profiling, a transcriptomic analysis, survival, protein–protein interaction networks, and a principal component analysis (PCA) were employed to elucidate functional associations and transcriptomic divergence. Subsequently, biomolecules from *A. javanica* were subjected to in silico screening using molecular docking and ADMET profiling. The docking protocol was validated using RK-33, a known *DDX3X* inhibitor. *DDX3X* was found to be linked to key leukemogenic pathways, including Wnt/β-catenin and MAPK signaling, indicating it to be a potential target. Molecular docking of *A. javanica* compounds revealed CIDs 15559724, 5490003, and 74819331 as potent *DDX3X* inhibitors with strong binding affinity and favorable pharmacokinetic and toxicity profiles compared to RK-33. This study highlights the importance of *DDX3X* in LAML pathogenesis and suggests targeting it using plant-derived inhibitors, which may require further in vitro and in vivo validation.

## 1. Introduction

Acute Myeloid Leukemia (LAML) is characterized by the rapid proliferation of abnormal myeloid cells in the bone marrow and peripheral blood and is a significant global health concern [1]. The Global Burden of Disease Study reported that new LAML cases rose from 79,372 in 1990 to 144,645 in 2021. Though the age-standardized incidence rate (ASIR) fell from 1.77 to 1.73 per 100,000 people during this time, the total number of cases grew because of population expansion and aging demographics. With Australia noting the highest rate of 4.9 per 100,000 people in 2021, high-income areas like North America and Western Europe have reported the highest ASIRs [2,3]. Leukemia is one of the most common cancers in Saudi Arabia. The Saudi Cancer Registry’s 2016 data showed an age-standardized rate of leukemia of 3.6 per 100,000 in men and 3.0 per 100,000 in women [4]. The highest incidence rates were reported by the Eastern Region. In particular, the average age at LAML diagnosis in Saudi Arabia seems lower than that in Western nations, implying possible regional or genetic elements affecting disease onset [4].

Due to its high morbidity and mortality rates, LAML requires fast diagnosis and treatment. In the United States, the five-year survival rate is roughly 30%; however, in Saudi Arabia, a study conducted at King Abdulaziz Medical City reported that the survival rate varies depending on cytogenetic and molecular abnormalities, underlining the need for individualized therapy [5,6].

The pathogenesis of LAML is accompanied by genetic mutations, where common mutations are fms-like tyrosine kinase 3 *(FLT3)* and Nucleophosmin 1 *(NPM1)* [7]. However, other mutations are being studied and are gaining attention. DEAD-box helicase 3 X-linked (*DDX3X)* is a frequently mutated gene in various cancers, including LAML [8]. The DEAD-box RNA helicases are a highly conserved family of ATP-dependent RNA-binding proteins that play key roles in various aspects of RNA metabolism, including transcription, splicing, export, translation, and decay [9]. Among these, *DDX3X* has emerged as a multifunctional protein. It is involved in the initiation of translation, particularly for mRNAs with long or highly structured 5′ untranslated regions [10]. Functionally, *DDX3X* contributes to RNA remodeling by unwinding secondary structures, regulating ribosome recruitment, and promoting RNA–protein complex fluidity in stress granules [11]. The DDX3 subfamilies contain N- and C-terminal tail extensions that harbor regulatory motifs essential for subcellular localization, protein–protein interactions, and oligomerization [12]. Despite considerable progress in characterizing the helicase core of *DDX3X*, the structural and functional dynamics of its flanking domains remain poorly understood, especially in the context of their contributions to disease pathology [13].

Recent studies have reported *DDX3X* as a recurrently mutated gene across various malignancies, including medulloblastoma, head and neck squamous cell carcinoma, natural killer/T-cell lymphoma, and non-small-cell lung cancer [14]. It is involved in genetically heterogeneous hematological malignancies with poor prognoses and limited targeted treatment options. This gene has been studied in solid tumors; however, its mechanistic role and the therapeutic potential for its inhibition in LAML remain underexplored [15]. *DDX3X* is a promising target for therapeutic interventions considering its essential role in RNA metabolism and its mutation-driven dysregulation in cancer; however, it has been underutilized as a therapeutic target [16]. In several preclinical cancer models, small-molecule inhibitors like RK-33 have been demonstrated as antiproliferative due to their targeting of the Walker A motif of *DDX3X* and impairment of its ATPase function. Even though RK-33 is capable of targeting *DDX3X*, it has the toxicity and pharmacokinetic constraints of synthetic inhibitors; therefore, this demands the investigation of more secure substitutes [17]. Thus, we selected *DDX3X* for our study to evaluate its druggability and therapeutic relevance, especially in the context of natural-product-based inhibition.

Natural products derived from medicinal plants offer a rich source of structurally diverse bioactive compounds with favorable safety profiles [18]. *Aerva javanica* (family: Amaranthaceae), a traditional medicinal plant widely used in Unani and Ayurvedic systems, is reputed for its anti-inflammatory, hepatoprotective, and anticancer properties [19]. *A. javanica* was selected for this study based on its traditional use in Unani and Ayurvedic medicine and its reported anticancer properties, making it a promising but understudied source of bioactive compounds targeting cancer-related proteins like DDX3X. Its phytoconstituents—comprising flavonoids, sterols, alkaloids, and terpenoids—have shown potential in modulating cancer-related pathways [20]. Despite its traditional significance, no prior study has examined the potential of *A. javanica* compounds for targeting RNA helicases such as *DDX3X*.

In this study, we conducted an integrated in silico investigation to evaluate the therapeutic relevance of *DDX3X* in LAML and to identify novel plant-derived inhibitors from *A. javanica*. By employing a combination of pan-cancer transcriptomics, mutational profiling, survival analyses, and molecular docking, we identified *DDX3X* as a highly mutated and differentially expressed gene in LAML. Subsequently, we screened *A. javanica* phytochemicals using ADMET and molecular docking approaches to identify compounds with favorable binding affinities within the ATP-binding pocket of *DDX3X*. The docking protocol was validated using the known *DDX3X* inhibitor RK-33. Our findings offer a promising framework for the development of plant-based *DDX3X* inhibitors and support the repositioning of *A. javanica* compounds as potential adjuncts in LAML therapy.

## 2. Results

### 2.1. Mutated Genes in Acute Myeloid Leukemia (LAML)

To identify key genes in LAML, we first analyzed the distribution of highly mutated genes using data from the GDC portal. The distribution of highly mutated genes in LAML was obtained through the GDC portal. B2M was found to be the mutated gene with the highest frequency, affecting over 13% of cases, followed by *DDX3X*, affecting ~7% of cases. Other genes, such as *ERBB3*, *BRCA1*, and *ASXL2*, are each mutated in approximately 4–5% of patients. Several other genes, including *NDRG1*, *ZNF3*, *CUX1*, and *ASXL1*, showed moderate frequencies, ranging between 2 and 3%, as shown in Figure 1. The identification of these mutated genes helps prioritize targets for functional studies, with *DDX3X* selected for this study due to its high mutation rate and its known oncogenic role in hematological malignancies [21].

### 2.2. The Expression Analysis of DDX3X

To assess the oncogenic relevance of *DDX3X* across malignancies and validate its potential as a target, we examined its expression profile in a pan-cancer context using TCGA data. The *DDX3X* expression varies by tumor type, with higher levels observed in DLBC, LAML, PAAD, and THYM and lower levels seen in KICH, PRAD, and LUAD, as shown in Figure 2. Notably, LAML showed the significant overexpression of *DDX3X*, indicating its potential role in leukemogenesis, which is consistent with previous research on *DDX3X’s* involvement in RNA metabolism and dysregulation in hematological malignancies [21].

### 2.3. Expression Profiling of DDX3X at Various Stages

Given the heterogeneity in the presentation of LAML, we next explored the DDX3X expression across disease subtypes, patient age groups, and racial categories to identify clinically relevant expression patterns. The expression profile of *DDX3X* in LAML shows significant variation by subtype, age, and race. In the subtype analysis, M0 patients exhibited the highest *DDX3X* levels, while M6 and M7 patients showed the lowest. The age-related expression was stable across most groups; however, a slight decrease in expression was observed in older patients aged 81–100. Racially, Asian patients were noted to have the highest expression, followed by Caucasians, with African Americans showing the lowest levels, suggesting potential ethnic regulatory influences despite the small sample sizes, as shown in Figure 3.

### 2.4. Survival Analysis of DDX3X

To evaluate the prognostic value of DDX3X in LAML, a survival analysis was performed to determine whether its expression levels correlated with overall and disease-free survival. The prognostic significance of *DDX3X* expression in LAML using Kaplan–Meier survival analyses was assessed using GEPIA 2. The results indicated no significant difference in the overall survival (OS) between patients with high and low *DDX3X* levels (log-rank *p* = 0.94; HR = 1.0), as shown in Figure 4. Additionally, the disease-free survival (DFS) analysis also showed no significant differences, with a log-rank *p*-value of 1.0 and an HR of 1.0, suggesting that *DDX3X* does not affect survival outcomes and that there is no association between its high expression and recurrence risk.

### 2.5. Principal Component Analysis (PCA) of DDX3X

To understand the transcriptomic impact of DDX3X and its associated genes further, we conducted a PCA to distinguish the expression profiles of LAML patients from those of healthy controls. To investigate the global transcriptional impact of *DDX3X* in LAML, the PCA was performed on genes positively correlated with *DDX3X* expression. The PCA showed a distinct separation between the LAML tumor samples and the whole blood controls, as shown in Figure 5. While the whole blood controls occupied a clearly separate area along the positive PC1 axis, the LAML samples clustered closely within the negative region of PC1. This significant separation emphasizes the possible involvement of DDX3X-related genes in leukemogenic transformation or disease-specific pathways in LAML by implying considerable transcriptomic variations.

### 2.6. Interactions of DDX3X with Other Proteins

A protein–protein interaction study was conducted using the STRING database to identify the possible molecular interactions and biological activities of DDX3X in LAML. The k-means algorithm grouped the resulting interaction network into three main groups, Cluster 1 (Green nodes), comprising proteins mostly involved in RNA splicing and processing, including DHX15, PRPF4B, and EIF1AX; Cluster 2 (Red nodes), comprising CCNT1, RLF, PAPOLA, and TMF1c, among others, with the proteins in this cluster mostly linked to post-transcriptional modification, RNA polymerase activity, and transcription control; and Cluster 3 (Blue nodes), containing proteins such as CTNNB1, APC, MAP2K4, and TAB3, which are key players in cell signaling, cytoskeletal organization, and apoptosis control, suggesting the possible involvement of DDX3X in signal transduction and cellular homeostasis, as shown in Figure 6.

### 2.7. A. javanica Compounds and Their Pharmacokinetic Screening 

To identify bioavailable and systemically active phytocompounds, we conducted a pharmacokinetic screening of *A. javanica*-derived molecules in comparison with the reference inhibitor RK-33. The pharmacokinetic profiling of the compounds from *A. javanica* showed a varied spectrum of absorption, distribution, and metabolism with respect to the control RK-33 (CID: 46184988). Several compounds, such as CID 15559724 and CID 5490003, showed significant P-gp inhibition, moderate BBB penetration, and high HIA values, suggesting their good oral bioavailability and systemic distribution. Their CYP3A4 inhibition potential, however, was lower than that of RK-33, suggesting less drug–drug interaction liability. However, the compounds CID 74978256 and CID 162998749 revealed low HIA and restricted membrane permeability, suggesting their lower absorption potential, as presented in Table 1.

The drug-likeness properties of *A. javanica* were evaluated against a set of thresholds using Lipinski’s Rule of Five. RK-33, the control compound, showed reasonable drug-likeness with no violations of Lipinski’s criteria. Compound 74819331 from *A. javanica* was found to pass the drug-likeness criteria, as shown in Table 2.

The toxicity profiles of the *A. javanica* compounds varied significantly. The control RK-33 was predicted to be active in the AMES test, was carcinogenic, and exhibited a very low LD, suggesting its moderately acute oral toxicity. The majority of the compounds from *A. javanica* were found to have low genetic toxicity, low hERG inhibition, low carcinogenicity, and a high LD50, indicating their low acute toxicity, as shown in Table 3.

### 2.8. Molecular Docking and Intra-Molecular Interaction Analysis of DDX3X and the A. javanica Compounds

To evaluate the binding efficiency and to propose the potential inhibitory activity of the selected phytocompounds against DDX3X, molecular docking was performed using iGEMDock. The reference inhibitor RK-33 (CID: 46184988) and the native substrate AMP (CID: 6083) served as control ligands. Although docking scores alone do not confirm inhibition, compounds that show favorable binding energies and interactions comparable to those for the known inhibitor RK-33 may serve as promising candidates for further experimental validation.

The compound CID 74978256 exhibited the lowest total binding energy, significantly better than that of RK-33 and AMP, suggesting its strong interaction affinity towards DDX3X. The compound also possessed a good van der Waals (VDW) contribution and moderate hydrogen bonding, indicating stable hydrophobic and hydrogen bonding interactions. Structural analysis of the compound uncovered several glycosidic moieties and phenolic hydroxyl groups that may help in hydrogen bonding and aromatic stacking. Another compound, CID 163189397, also exhibited a better binding energy than that of the control and the substrate and was involved in multiple favorable interactions, such as an attractive charge, conventional hydrogen bonds, and pi-donor hydrogen bonds, especially involving the key residues THR201, ARG202, and HIS527. These residues also interact with RK-33 and AMP, suggesting potential competitive inhibition. Another potent compound, CID 74819331, was found to form pi–cation, pi–anion, and conventional hydrogen bonds with the active site residues ARG252, CYS317, and GLU256, reflecting its favorable electrostatic and hydrophobic complementarity. However, the control, RK-33, the known inhibitor, formed carbon–hydrogen bonds and pi–pi T-shaped interactions primarily with the residues HIS227, PRO203, and THR201, indicating its more specific yet weaker interactions than those of several of the phytocompounds. The top-binding compounds 74978256, 163189397, and 74819331 possess large polyphenolic structures with aromatic rings, hydroxyl groups, and sugar moieties. These structures enable pi–alkyl interactions with key aromatic residues. CID 74978256, for instance, has several aromatic rings and sugar chains, assisting in the formation of various types of interactions. In contrast, RK-33 and AMP are more basic in structure, with no significant H-bond donor/acceptor capacity or steric binding pocket coverage. Though polar, AMP lacks notable hydrophobic interactions that many phytocompounds attain via aromatic scaffolds. This study shows that many active site residues are shared in the top compounds, including ARG202 and THR201, which create hydrogen bonds or pi interactions; HIS527 and TYR200, which serve as anchoring points for pi–pi stacking or T-shaped interactions; PRO205 and GLY229, which act as contact points for carbon–hydrogen or pi–alkyl interactions; and GLU285 and ARG531, which participate in electrostatic (pi-anion or attractive charge) interactions, as shown in Table 4 and Figure 7.

## 3. Discussions

To explore the role of *DDX3X* in LAML, highly mutated genes in LAML were identified, and B2M was found as the most frequently mutated gene. However, the current study focused on the second highest mutated gene, *DDX3X*, due to its underexplored role in LAML. Previous studies have revealed a link of *DDX3X* to both oncogenic and tumor-suppressive pathways, making it an interesting candidate for functional validation and therapeutic investigation [15]. Therefore, molecular docking experiments, co-expression profiling, and a survival analysis were given top priority for this gene, using the known inhibitor RK-33 as a reference. The current study found this gene to be upregulated in LAML, which aligns with earlier reports of *DDX3X* functioning as an RNA helicase that promotes oncogenic transformation by regulating mRNA translation, cell cycle progression, and apoptosis evasion [22]. Its significantly elevated expression in LAML supports its selection as a candidate for further functional and therapeutic exploration. However, our findings reveal that *DDX3X* is not always upregulated across all cancers; in fact, it was found to be downregulated in solid tumors like PRAD and LUAD, implying its potential tumor-suppressive role. This dual nature aligns with prior findings, where *DDX3X* can function both as an oncogene and a tumor suppressor depending on the cellular and molecular context. Therefore, these findings provided a strong basis for selecting *DDX3X* in LAML for in-depth studies, including a survival correlation, a pathway analysis, and therapeutic targeting via small-molecule inhibitors.

Our study on *DDX3X* found significant variations linked to biological and demographic factors by employing UALCAN. Notably, higher levels of *DDX3X* were observed in the M0 and M5 subtypes, suggesting its role in early hematopoietic progenitors and monocytic lineages, which aligns with its functions in RNA helicase activity and cell differentiation. Its expression was observed to be stable across age groups; however, a slight decline in the oldest cohort was noted. This decline hints at potential age-related epigenetic changes or clonal selection in leukemic progression. Additionally, racial disparities emerged, with elevated *DDX3X* levels in Asians compared to lower levels in African Americans, prompting further investigation into the genetic and environmental influences on its regulation. Therefore, these findings emphasize the need for stratified approaches in future research and therapeutic strategies for exploring *DDX3X* in LAML.

Our findings revealed that variations in *DDX3X* levels did not significantly influence OS or DFS outcomes. The survival analysis, which included Kaplan–Meier curves and non-significant log-rank *p*-values, showed that both high- and low-*DDX3X*-expression groups displayed comparable survival patterns, implying that *DDX3X* is not a prognostic biomarker in LAML. However, earlier studies have suggested its important role as a tumor suppressor or as an oncogene in various cancers; therefore, our findings draw attention to further investigations into its molecular mechanisms and possible functions in treatment responses and disease progression. This could be achieved by means of functional assays and larger patient populations.

The PCA revealed a clear separation between LAML and normal whole blood samples depending on the *DDX3X*-correlated gene expression. This implies that *DDX3X* could be essential to a particular gene network affecting leukemic cell behavior, therefore distinguishing it from healthy hematopoietic conditions. Although the survival analysis did not identify *DDX3X* as a prognostic factor, the observed transcriptomic divergence suggests its possible use as a diagnostic tool or for greater knowledge of LAML’s mechanisms. Genes linked to *DDX3X* might participate in important cellular processes, including proliferation, RNA metabolism, or immune evasion, particularly in leukemic transformation.

The protein–protein interaction analysis highlighted the integral role of DDX3X in LAML. It was observed to interact with RNA-binding proteins and splicing regulators, underscoring its function in RNA helicase activity and mRNA export. DDX3X is associated with transcriptional co-regulators and RNA polymerase II elongation factors, suggesting its role in regulating gene expression, which may influence leukemic cell differentiation and proliferation. Its interactions with CTNNB1 (β-catenin) and MAP kinase components indicate a regulatory function in the Wnt/β-catenin and MAPK signaling pathways, which are frequently dysregulated in leukemia. Overall, the protein–protein interaction analysis indicates DDX3X as a regulatory hub in leukemia, positioning it as a prospective therapeutic target.

This study focused on identifying potential DDX3X inhibitors from *A. javanica* compounds through molecular docking and ADMET profiling. The compounds CID 15559724 and CID 5490003 exhibited high HIA, moderate Caco-2 permeability, and significant P-gp inhibition, indicating their favorable oral bioavailability and a reduced risk of drug–drug interactions compared to those for the reference drug, RK-33, which showed higher CYP3A4 inhibition and a lower safety profile. CID 74819331 was highlighted as a viable oral drug candidate, meeting Lipinski’s Rule of Five with a balanced pharmacokinetic profile, moderate HIA, and acceptable BBB penetration. The compounds CID 15559724 and 5490003 share a remarkably similar core scaffold. Both exhibit a methoxy-substituted aromatic ring (COc1ccc-) linked through a conjugated system (-c2cc(=O)c3c(O)cc(OC)c-), suggesting that they belong to the same class of flavonoid or phenolic derivatives. The presence of hydroxyl (-OH) and methoxy (-OCH₃) groups in both molecules contributes to their polarity and potential for hydrogen bonding, which may enhance their binding affinity toward protein targets such as DDX3X [23,24]. However, CID 74819331 is characterized by a fused bicyclic system with a lactone (O = C1CC…) and multiple hydroxylated phenyl rings. It also contains a sugar moiety (C1OC(CO)C(O)C(…)), suggesting a glycosylated derivative, making it a bulkier and more hydrophilic architecture, which significantly increases its molecular weight and hydrophilicity. This structural difference implies divergent pharmacokinetic and pharmacodynamic behavior compared to that of the other two. CID 74819331 exhibited the lowest binding energy (−147.7 kcal/mol), which may result from its multiple interaction types, such as conventional hydrogen bonds, pi–cation interactions, pi–anion interactions, and pi–alkyl contacts, enhancing its binding specificity. The combination of polar and non-polar interactions across different regions of the DDX3X active site suggests that 74819331 achieves extensive, multipoint anchoring, resulting in the stronger affinity. CIDs 5490003 and 15559724 displayed similar binding energies (~−120 kcal/mol), with hydrogen bonds with key residues, carbon–hydrogen and Pi–Pi interactions, and pi–alkyl and Pi–sigma interactions contributing to hydrophobic stabilization. The slight energy difference (~1.5 kcal/mol) is likely due to minor steric or electronic differences.

In contrast, compounds CID 74978256 and CID 162998749 demonstrated poor absorption characteristics and significant violations of the drug-likeness parameters. Toxicity assessments revealed that most of the *A. javanica* compounds had low mutagenicity and carcinogenicity risks, with higher LD_50_ values, while CID 74819331 showed AMES positivity and a lower LD_50_, necessitating further validation. CID 15559724 and CID 5490003 were found to be promising hits due to their excellent binding affinity and acceptable ADMET behavior, though they did not fully comply with the drug-likeness criteria. CID 74819331, although slightly less potent in binding, emerged as a well-balanced lead molecule with good docking, drug-likeness, and safety features. Future studies should focus on structural optimization of these candidates to improve their drug-likeness and pharmacokinetic properties, with the recommendation of additional in vitro and in vivo evaluations to confirm their pharmacological efficacy and safety.

## 4. Materials and Methods

### 4.1. The Data Sources and Bioinformatics Tools

In this study, we employed a combination of bioinformatics platforms and publicly available databases to investigate *DDX3X* expression patterns and their potential prognostic value in various human cancers. The retrieved data were analyzed using the GDC Data Portal (https://portal.gdc.cancer.gov/ (accessed on 14 October 2024)) for access to raw and processed genomic datasets [25]; GEPIA (Gene Expression Profiling Interactive Analysis) (http://gepia.cancer-pku.cn (accessed on 14 October 2024)), which integrates TCGA and GTEx expression data, to analyze the differential gene expression between tumor and normal tissues [26]; and UALCAN (http://ualcan.path.uab.edu (accessed on 14 October 2024)) to analyze the expression levels of *DDX3X* across different pathological stages using TCGA-derived datasets [27].

### 4.2. The Identification of Highly Mutated Genes in Acute Myeloid Leukemia (LAML)

The Cancer Genome Atlas (TCGA) (https://portal.gdc.cancer.gov/ (accessed on 15 October 2024)) was accessed via the GDC Data Portal to obtain the mutation profiling for LAML. In this study, the gene with the highest mutation rate (*B2M*) was initially identified and excluded from further analysis due to its extensive prior characterization and the in-depth exploration of its oncogenic role and therapeutic implications in the existing literature [28,29]. Instead, the gene with the second highest mutation frequency, *DDX3X*, was selected for focused investigation, as its function and clinical relevance in LAML remained less explored.

### 4.3. The Pan-Cancer Expression Analysis

The GEPIA 2 tool (http://gepia2.cancer-pku.cn/ (accessed on 16 October 2024)) was used to analyze the expression of *DDX3X* mRNA across 27 cancer types. This tool allowed for the integration of TCGA tumor data with normal tissue data from the GTEx project, ensuring more robust comparisons. We applied the following parameters: log2(TPM + 1) transformation, an ANOVA for differential expression, a |log2 fold-change| ≥ 1, and a q-value < 0.01. The tumor types examined included ACC, BLCA, BRCA, COAD, DLBC, ESCA, GBM, HNSC, KICH, KIRC, KIRP, LAML, LGG, LIHC, LUAD, LUSC, OV, PAAD, PRAD, READ, SKCM, STAD, TGCT, THCA, THYM, UCEC, and UCS [30].

### 4.4. Stage-Specific Expression Profiling

UALCAN (https://ualcan.path.uab.edu/index.html (accessed on 16 October 2024)) was used to examine the variation in the *DDX3X* expression across different clinical stages (Stages I–IV) in the LAML patients in the TCGA dataset. This tool provided stratified gene expression profiles based on pathological staging.

### 4.5. Survival Analysis

To understand the prognostic relevance of DDX3X in LAML, survival outcomes associated with DDX3X expression were assessed using GEPIA 2. The Kaplan–Meier method was employed to generate the overall survival (OS) [31] and disease-free survival (DFS) curves. The log-rank test was used to evaluate statistical significance, and hazard ratios (HRs) were calculated using the Cox proportional hazards model. A median split (50%) was applied to defining the high- and low-expression groups for *DDX3X*.

### 4.6. The Co-Expression Analysis

To identify genes that were transcriptionally linked to *DDX3X*, a co-expression analysis was conducted to explore regulatory mechanisms or participation in similar biological processes using the R2 Genomics Analysis and Visualization Platform (https://hgserver1.amc.nl/cgi-bin/r2/main.cgi?open_page=login) (accessed on 17 October 2024) [32].

### 4.7. The Principal Component Analysis (PCA)

To evaluate the transcriptional differences between the LAML tumor samples and normal whole blood regarding DDX3X expression, a PCA was conducted using the GEPIA 2 web server. Genes with a strong positive correlation with *DDX3X* (a Pearson’s coefficient > 0.6) were identified and analyzed through the PCA. The analysis utilized transcriptomic data from TCGA-LAML tumor samples and GTEx whole blood controls, resulting in a PCA plot that illustrated the first two principal components (PC1 and PC2).

### 4.8. The Protein–Protein Interaction Network

To explore the protein–protein interactions of DDX3X with its associated genes, the top 20 positively correlated genes were submitted to the STRING v11.5 (https://string-db.org/ (accessed on 17 October 2024)) database [33]. Functional and physical associations were visualized based on confidence scores derived from multiple types of evidence, including experimental data, databases, and co-expression.

### 4.9. Pharmacokinetic Screening of A. javanica Compounds

The 3D and 2D structures of the phytochemicals derived from *A. javanica* were obtained via PubChem and assessed for drug-likeness using ADMETlab 2.0 (https://admetmesh.scbdd.com/ (accessed on 22 October 2024)) [17]. This tool predicts absorption, distribution, metabolism, excretion, and toxicity (ADMET) parameters. Compounds that met the pharmacokinetic and safety criteria were shortlisted for molecular docking studies.

### 4.10. The Molecular Docking Analysis

The crystal structure of DDX3X (UniProt ID: O00571) was retrieved from the Protein Data Bank (PDB). The X-ray-determined 3D structure resolved at 2.2 Å, bound with the substrate adenosine monophosphate (PDB ID: 2I4I_A) [34], was processed to remove water molecules and bound ligands, correct structural inconsistencies, and prepare it for docking using iGEMDOCK (http://gemdock.life.nctu.edu.tw/dock/igemdock.php (accessed on 26 October 2024)) [15]. Subsequently, the phytochemicals from *A. javanica* were processed and geometrically optimized to dock into the binding pocket of 2I4I, within the Walker I motif that contributes to ATP binding [35], and where the inhibitor RK-33 binds [36].

### 4.11. Interaction Profiling and Visualization

The docked complexes were visualized using Discovery Studio Visualizer v21 and PyMol v2.5. These tools enabled a detailed examination of the ligand-binding modes, key interacting residues, and the pharmacophoric features essential for activity.

### 4.12. Validation of the Docking Protocol Using RK-33

To validate the reliability and accuracy of the molecular docking protocol used in this study, the known DDX3X inhibitor RK-33 was docked into the active binding site of the DDX3X crystal structure (PDB ID: 2I4I). RK-33 is a small-molecule inhibitor that effectively targets DDX3X helicase activity and shows promise as an anticancer agent [37]. Docking was performed using consistent parameters, and the binding pose of RK-33 was successfully reproduced as a positive control. This validated the iGEMDOCK software’s v 2.1 ability to predict relevant interactions in the DDX3X binding pocket, which increased the confidence in the docking results for the *A. javanica* -derived compounds.

## 5. Conclusions

In this study, the role of *DDX3X* in LAML was explored, focusing on its mutational, transcriptomic, and therapeutic relevance. Our findings substantiate its elevated expression in LAML relative to that in the normal hematopoietic tissue. The PCA indicates that *DDX3X* may have diagnostic or mechanistic importance. *DDX3X* was found to be involved in critical cellular pathways, including Wnt/β-catenin and MAPK signaling, both of which are frequently dysregulated in leukemia. All of these findings strengthen the rationale for targeting *DDX3X* in therapeutic design. Therefore, compounds from A. javanica, a traditionally used medicinal plant, were identified as novel *DDX3X* inhibitors. This integrated computational analysis reveals that while structurally similar compounds (5490003, 15559724) show consistent and promising interactions with *DDX3X*, structurally distinct molecules like 74819331 can also provide superior binding energies. Together, these findings highlight the chemical diversity of *A. javanica* and its potential as a source of novel *DDX3X* modulators. Compared to RK-33, a synthetic *DDX3X* inhibitor with known pharmacokinetic limitations, these plant-derived compounds offered superior safety profiles and a lower risk of drug–drug interactions. Our integrated in silico approach underscores *DDX3X* as a functionally significant but non-prognostic marker in LAML and highlights the therapeutic promise of A. javanica-derived compounds. This work paves the way for future preclinical and functional validation of these bioactive molecules, establishing a foundation for plant-based therapeutic strategies targeting RNA helicases in hematological malignancies. Further experimental studies may be required to validate these computational insights and investigate the mechanistic effects of selected compounds on *DDX3X* activity and leukemic cell viability.

## Figures and Tables

**Figure 1 ijms-26-05445-f001:**
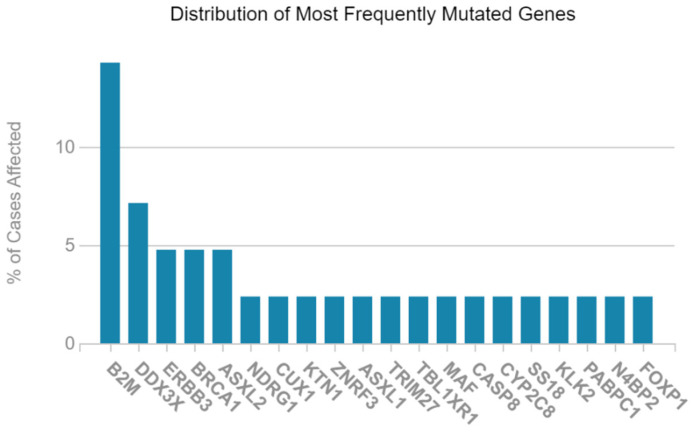
The distribution of the most frequently mutated genes in Acute Myeloid Leukemia (LAML). The bar graph illustrates the percentage of LAML cases affected by mutations in the top mutated genes based on TCGA data. The gene B2M exhibits the highest mutation frequency (>13%), followed by DEAD-box helicase 3 X-linked (*DDX3X*), human epidermal growth factor receptor 3 (*ERBB3*), breast cancer gene 1 (*BRCA1*), and ASXL Transcriptional Regulator 2 (*ASXL2*), with each affecting 4–8% of cases.

**Figure 2 ijms-26-05445-f002:**
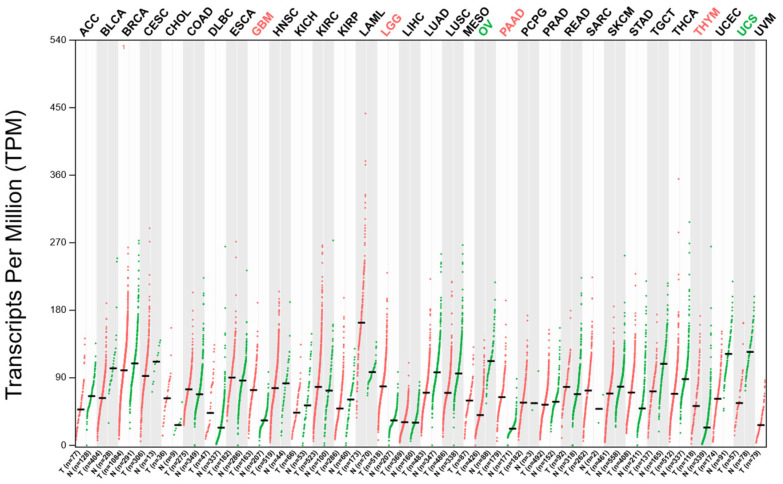
The pan-cancer expression profile of *DDX3X* was analyzed across TCGA tumor datasets, comparing the transcript levels (TPM) between tumor (red) and normal (green) tissues for various cancer types using GEPIA 2 data. Each dot indicates the individual sample expression, with the black lines showing the median expression levels.

**Figure 3 ijms-26-05445-f003:**
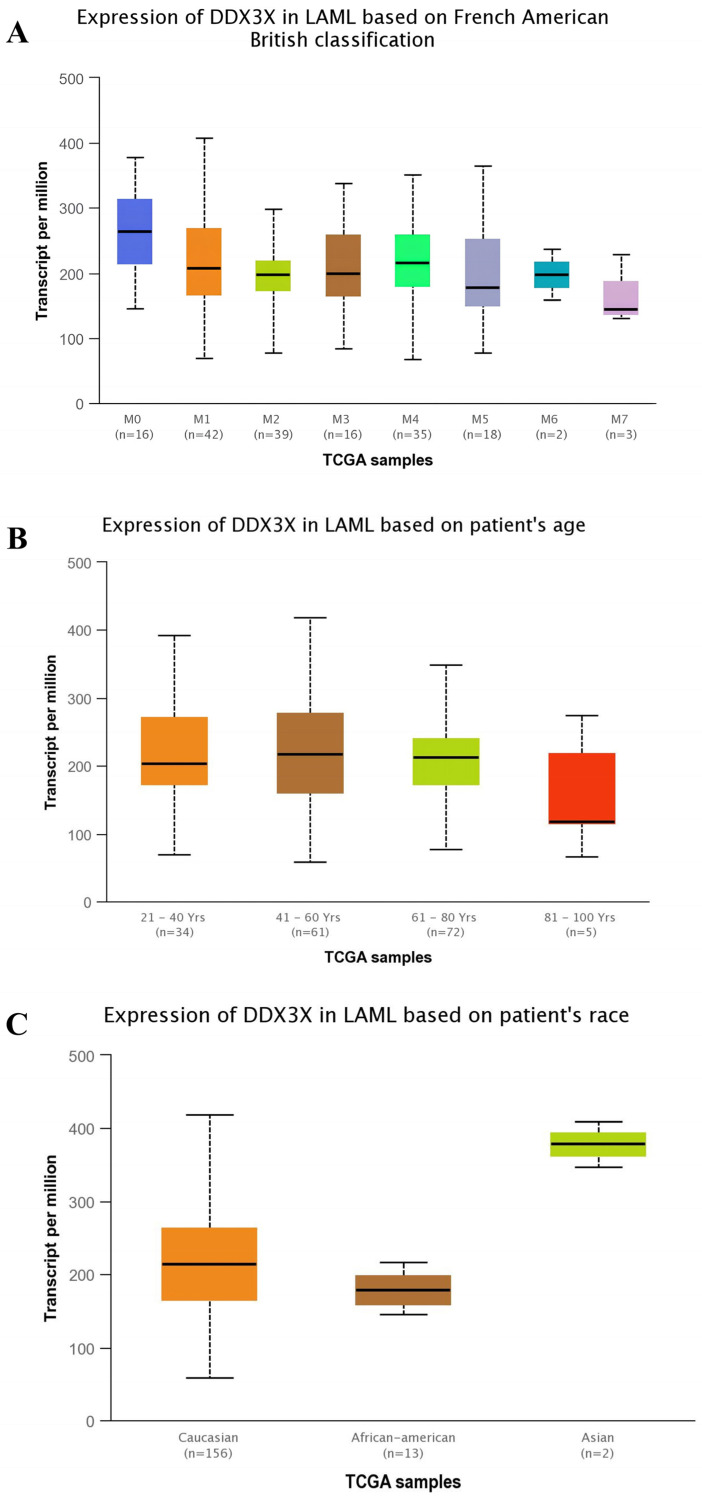
The expression analysis of DDX3X in Acute Myeloid Leukemia (LAML). (**A**) The DDX3X expression across different LAML subtypes (M0–M7) based on the French–American–British (FAB) classification. (**B**) The expression in LAML patients grouped by age ranges (21–100 years). (**C**) The expression among LAML patients from different racial backgrounds. Transcript levels are measured in Transcripts Per Million (TPM). Box plots show the medians, interquartile ranges, and overall distribution.

**Figure 4 ijms-26-05445-f004:**
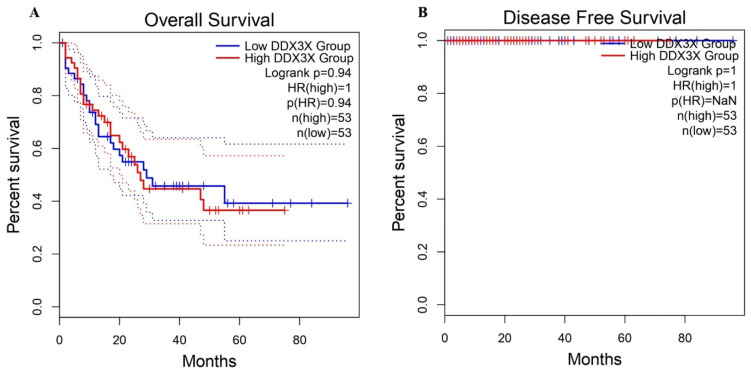
The survival analysis of the DDX3X expression in patients with LAML. (**A**) The Kaplan–Meier curve for the overall survival (OS) comparing patients with high and low DDX3X expression levels. The *p*-value of 0.94, with a hazard ratio (HR) of 1.0, indicates no significant difference in the OS between groups. (**B**) The disease-free survival (DFS) analysis also demonstrated no difference between the high- and low-DDX3X-expression groups.

**Figure 5 ijms-26-05445-f005:**
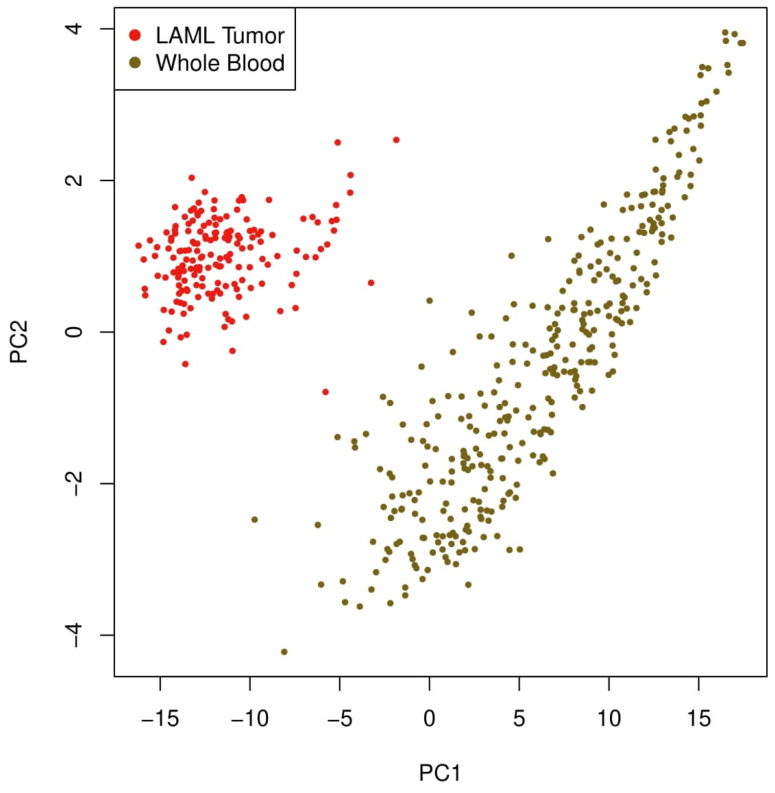
The principal component analysis (PCA) plot of genes positively correlated with *DDX3X* in LAML. The PCA was performed to assess the expression pattern of *DDX3X*-correlated genes in LAML tumor samples (red) versus whole blood controls (brown), indicating distinct variance in LAML vs. whole blood controls.

**Figure 6 ijms-26-05445-f006:**
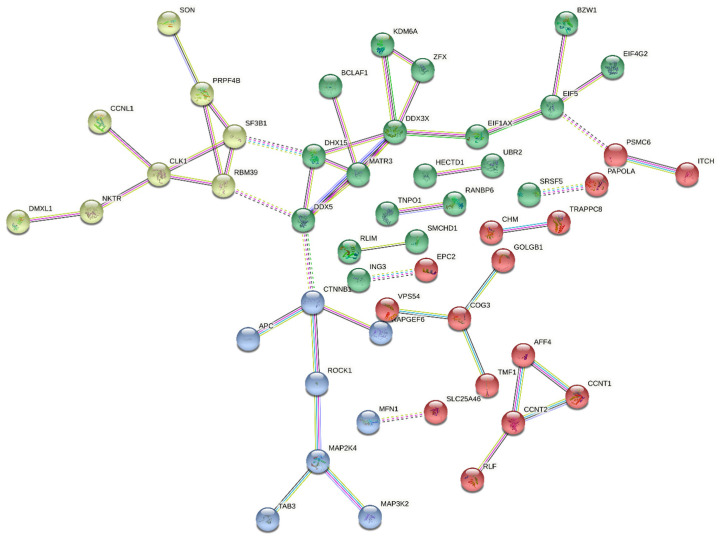
The DDX3X protein network. Proteins with a strong positive correlation with DDX3X in LAML are clustered into three categories based on k-means clustering in the STRING software v.12.0.

**Figure 7 ijms-26-05445-f007:**
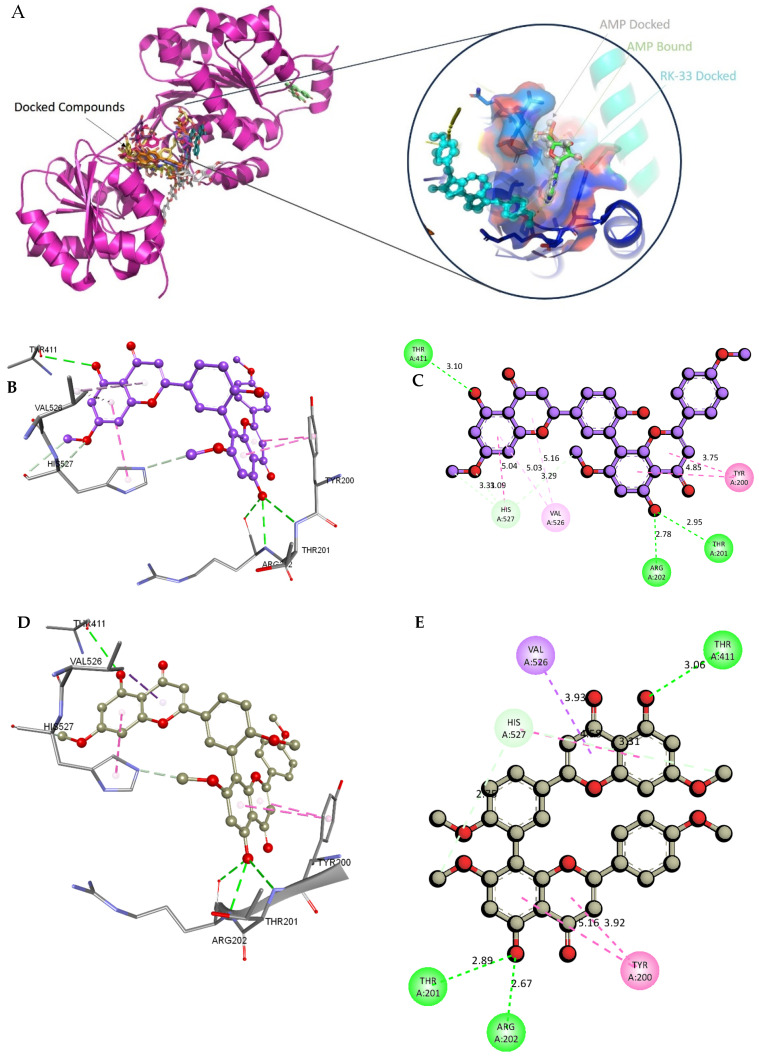
Docking and intra-molecular interactions. (**A**) All of the superimposed *A. javanica* compounds (as stick representations) docked into the binding pocket of DDX3X (ODB ID: 2I4I_A) (as cartoon representations), with the top compounds from *A. javanica* shown in 3D to show the binding surface of the receptor and in 2D representation to show the intra-molecular interactions and their distances in Å. Green dashes represent hydrogen bonds, pink dashes represent hydrophobic interactions, and orange dashes represent electrostatic interactions. Docked compounds: CID 15559724, (**B**) 3D and (**C**) 2D; CID 5490003, (**D**) 3D and (**E**) 2D; CID 74819331, (**F**) 3D and (**G**) 2D.

**Table 1 ijms-26-05445-t001:** The computational assessment of the pharmacokinetic properties of *A. javanica* compounds.

CID	HIA	Caco-2 Permeability	Pgp Inhibitor	BBB Penetration	PPB	CYP3A4 Inhibitor
5281643	0.720165	0.024945	0.019489	0.36468	0.34701	0.005815
5282149	0.813789	0.04693	0.015533	0.52433	0.3447	0.0056
5320646	0.834688	0.038304	0.779624	0.070213	1.064644	0.102586
5490003	0.915642	0.06874	0.943979	0.364681	0.970795	0.243789
15559724	0.916661	0.067603	0.902655	0.229925	1.035477	0.287866
74819331	0.720941	0.06123	0.028965	0.681398	0.402518	0.084012
74978256	0.296633	0.005106	0.438269	0.160238	0.878509	0.270137
91885208	0.764231	0.033519	0.019925	0.370717	0.480753	0.00434
162817595	0.576042	0.026792	0.874571	0.034886	1.014972	0.088901
162817597	0.200472	0.003026	0.020575	0.1995	0.375597	0.011781
162998749	0.376207	0.006379	0.022332	0.161656	0.399283	0.004699
163189397	0.381535	0.010827	0.895205	0.02035	0.891346	0.06044
46184988	0.97246	0.772927	0.321354	0.894446	0.879919	0.849846
6083	0.133681	0.006179	0.019417	0.272034	0.157684	0.008052

Abbreviations: HIA: human intestinal absorption; Caco-2 permeability: Caco-2 cell monolayer permeability; Pgp inhibitor: P-glycoprotein inhibitor; BBB: blood–brain barrier; PPB: plasma protein binding; CYP3A4 inhibitor: Cytochrome P450 3A4 Enzyme inhibitor.

**Table 2 ijms-26-05445-t002:** The drug-likeness properties of selected *A. javanica* compounds.

CID	MW	LogP(o/w)	TPSA	HBAs	HBDs	Lipinski Violations
5281643	464.379	−0.62847	210.51	12	8	Not accepted
5282149	448.38	−0.63566	190.28	11	7	Not accepted
5320646	566.518	5.267783	159.8	10	4	Not accepted
5490003	594.572	5.576282	137.8	10	2	Not accepted
15559724	580.545	5.608342	148.8	10	3	Not accepted
74819331	418.398	−0.43163	156.91	9	6	Accepted
74978256	770.693	1.493971	284.73	18	9	Not accepted
91885208	478.406	−0.53838	199.51	12	7	Not accepted
162817595	624.551	2.556964	225.81	14	7	Not accepted
162817597	624.548	−1.07282	258.43	16	9	Not accepted
162998749	552.485	−1.25717	250.97	14	10	Not accepted
163189397	870.81	2.335287	289.03	20	7	Not accepted
46184988	428.452	2.046065	96.95	9	0	Accepted
6083	347.224	−2.69664	186.07	10	5	Accepted

Abbreviations: MW—Molecular Weight; LogP(o/w)—Logarithm of the Partition Coefficient (octanol/water); TPSA—Topological Polar Surface Area; HBAs—Hydrogen Bond Acceptors; HBDs—Hydrogen Bond Donors.

**Table 3 ijms-26-05445-t003:** In silico toxicity predictions for *A. javanica* compounds.

CID	AMES Toxicity	Mutagenicity	Carcinogens	Hepatotoxicity	hERG Inhibitor	LD50 (mg/kg)
5281643	0.777389	Inactive	0.627358	Inactive	0.015333	5000
5282149	0.848903	Inactive	0.684536	Inactive	0.008304	5000
5320646	0.640871	Inactive	0.672932	Inactive	0.1671	4000
5490003	0.836254	Inactive	0.551992	Inactive	0.317555	4000
15559724	0.702942	Inactive	0.556323	Inactive	0.255199	4000
74819331	0.326982	Active	0.456241	Inactive	0.008861	2000
74978256	0.210756	Inactive	0.25206	Inactive	0.088186	5000
91885208	0.779684	Inactive	0.589165	Inactive	0.00742	5000
162817595	0.625209	Active	0.40003	Inactive	0.170601	5000
162817597	0.27347	Inactive	0.284445	Inactive	0.048891	5000
162998749	0.533229	Inactive	0.401097	Inactive	0.00953	5000
163189397	0.50833	Inactive	0.232467	Inactive	0.115662	5000
46184988	0.507371	Active	0.621935	Inactive	0.004847	360
6083	0.289806	Inactive	0.286742	Inactive	0.074652	11,250

**Table 4 ijms-26-05445-t004:** The binding energies and key intermolecular interactions of *A. javanica* compounds with DDX3X (PDB ID: 2I4I_A) compared to the reference inhibitor RK-33 and the native substrate AMP.

CID	Two-Dimensional Structure	vDW	Binding Energy	Intra-Molecular Interactions
RK-33 Docked	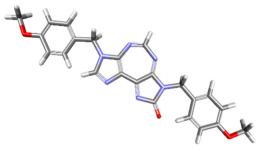	−81.6631	−94.781	Carbon–Hydrogen Bonds: HIS227, PRO203, THR201. Pi–Pi T-shaped: HIS527. Alkyl: ARG202. Pi–Alkyl: PRO205, ARG202.
74978256	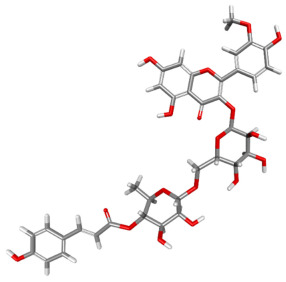	−135.631	−159.906	Conventional Hydrogen Bonds: TYR200, THR201, GLN207. Carbon–Hydrogen Bonds: GLY229, TYR200. Pi–Anion: GLU285, TYR200. Pi-Donor Hydrogen Bonds: THR201, ARG202. Pi–Sigma: TYR200. Pi–Alkyl: ARG202.
163189397	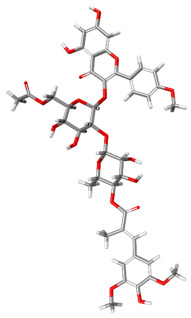	−128.61	−155.06	Attractive Charge: ARG202, GLU524. Conventional Hydrogen Bonds: THR201, ARG202. Carbon Hydrogen Bonds: SER520, THR201. Pi-Donor Hydrogen Bonds: THR202. Pi–Sigma: THR201. Pi–Pi T-Shaped: HIS527. Pi–Alkyl: ARG202.
74819331	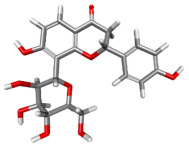	−129.001	−147.727	Conventional Hydrogen Bonds: ARG252, CYS317. Pi–Cation: ARG262. Pi–Anion: GLU256. Pi–Alkyl: ARG252.
5490003	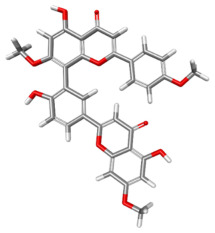	−107.721	−120.524	Conventional Hydrogen Bonds: THR201, ARG202, THR411. Carbon–Hydrogen Bonds: HIS527. Pi–Sigma: VAL526. Pi–Pi Stacked: TYR200. Pi–Pi T-Shaped: HIS527.
15559724	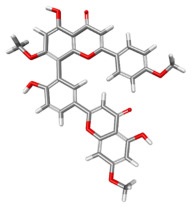	−107.546	−119.038	Conventional Hydrogen Bonds: THR201, ARG202, THR411. Carbon Hydrogen Bonds: HIS527. Pi–Pi Stacked: TYR200. Pi–Pi T-Shaped: HIS527. Pi–Alkyl: VAL526.
162817595	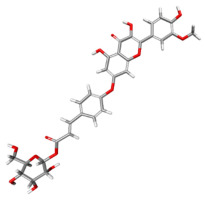	−97.1523	−116.779	Conventional Hydrogen Bonds: GLY227, GLY229, LYS230, VAL526. Carbon Hydrogen Bonds: GLY530, GLN281, SER228. Pi–Pi Stacked: HIS527. Alkyl: VAL405. Pi–Alkyl: PRO205.
162817597	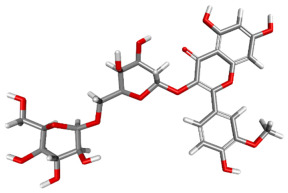	−73.4756	−115.757	Conventional Hydrogen Bonds: THR411, SER412, GLY504, ARG531, ALA502, LEU505, GLY530. Carbon–Hydrogen Bonds: MET167. Amide–Pi Stacked: GKY227. P–Akyl: PRO205.
AMP Docked	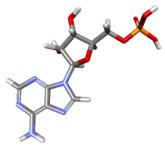	−77.5459	−115.411	Conventional Hydrogen Bonds: TYR200, GLN207, GLY227, SER228, GLY229, LYS230, THR231, ARG202. Pi–Pi Stacked: TYR200. Pi–Pi Alkyl: ALA232. Attractive Charges: LYS230.
5281643	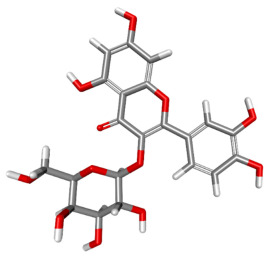	−72.3283	−112.611	Conventional Hydrogen Bonds: GLY504, ARG531, PRO203, ALA502, LEU505, GLY227. Carbon–Hydrogen Bonds: THR204, PRO205, ARG503. Pi–Cation: MET167. Pi–Pi Stacked: HIS527. Pi–Alkyl: PRO205.
5320646	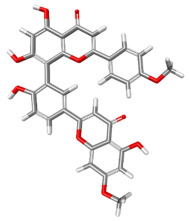	−87.5561	−112.148	Conventional Hydrogen Bonds: ARG199, THE201, GLN207, LYS288, ARG199, ARG202, GLN207. Pi–Sigma: ARG199. Pi–Pi Stacked: TYR200. Alkyl: LYS288. Pi–Alkyl: ARG199.
5282149	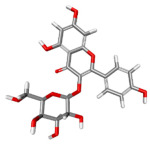	−68.5132	−108.82	Conventional Hydrogen Bonds: LYS208, GLY504, HIS527, ARG531LEU505. Pi–Alkyl: PRO205, ARG503.
91885208	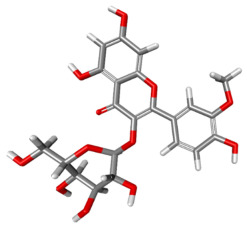	−73.5348	−106.394	Conventional Hydrogen Bonds: VAL206, SER228, THR411, ARG531, ARG534, THR226, GLY530, LEU505, SER228. Alkyl: VAL168, PRO205, VAL206, VAL405. Pi–Alkyl: PRO205.

## Data Availability

The data can be made available on request.

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
