# Peer review of "Repurposing Biomolecules from Aerva javanica Against DDX3X in LAML: A Computer-Aided Therapeutic Approach"

_ijms, 2025, doi:10.3390/ijms26125445_

Round 1

Reviewer 1 Report

Comments and Suggestions for Authors

The authors embarked on characterising natural products as novel inhibitors of DDX3(X) via a in silico workflow. The hypothesis and workflows are sound and the experimental procedures alllow for investigation of the aims. However, the authors should;

(1) already explain in the Results section why DDX3(X) was picked over B2M.

(2) expand on the results section more, as it is difficult for the reader to follow why a particular analysis was performed. I appreciate that the explanation was picked up in the Discussion, however, a proper introduction to the individual experiments will significantly increase readability and understanding of the research done. 

Minor issues:

  • Please maintain a consistent writing style for latin names and/or in silico. In multiple instances A. javanica was not formatted in italic.
  • In table 4 it would make sense to only have one entry for AMP.

Author Response

The authors embarked on characterising natural products as novel inhibitors of DDX3(X) via a in silico workflow. The hypothesis and workflows are sound and the experimental procedures alllow for investigation of the aims. However, the authors should;

Comment 1: already explain in the Results section why DDX3(X) was picked over B2M.

Response: We thank the reviewer for the insightful comment. In the revised version of the Introduction, we have now clearly explained the justification for choosing DDX3X over B2M, despite B2M having a higher mutation frequency.

Comment 2: expand on the results section more, as it is difficult for the reader to follow why a particular analysis was performed. I appreciate that the explanation was picked up in the Discussion, however, a proper introduction to the individual experiments will significantly increase readability and understanding of the research done. 

 Response: we are thankful to the reviewer. In the revised version we have updated with an introductory line to the results section to describe the relevance of the investigation.

Minor issues:

Comment 3: Please maintain a consistent writing style for latin names and/or in silico. In multiple instances A. javanica was not formatted in italic.

Response: In the revised version of the manuscript, we have updated the scientific name in italics.

Comment 4: In table 4 it would make sense to only have one entry for AMP.

Response: Thank you for the advice. We have updated the Table 4.

Reviewer 2 Report

Comments and Suggestions for Authors

Introduction Conciseness: The introduction would benefit from condensation. Sentences such as "Its incidence and impact vary by region, with especially significant consequences for global healthcare systems" and "The overall survival rates are found to be low among older people, despite treatment developments," appear redundant and could be rephrased or omitted for brevity.

Mutational Analysis: The study effectively underscores the role of DDX3X in LAML. To enhance the findings, the authors might consider conducting a mutational analysis for DDX3X, akin to the analysis reported for FLT3 and NPM1.

Plant Name Formatting: Ensure consistent italicization of all plant names throughout the manuscript.

Rationale for Compound Selection: The authors should elucidate their rationale for selecting natural compounds over synthetic ones for this study.

Justification for Aerva javanica Selection: Please provide a brief explanation in the introduction regarding the selection of Aerva javanica for this study.

Exclusion of B2M: Figure 1 indicates that B2M is the most frequently mutated gene in LAML. Please clarify why this gene was excluded from the current study's scope.

Figure 3 Resolution: A high-resolution image for Figure 3 is necessary for publication.

Tool URLs: Please provide URLs for all computational tools employed in the study.

In Vitro Validation: The authors may consider conducting in vitro validation studies to further substantiate their findings.

Author Response

Comment 1: Introduction Conciseness: The introduction would benefit from condensation. Sentences such as "Its incidence and impact vary by region, with especially significant consequences for global healthcare systems" and "The overall survival rates are found to be low among older people, despite treatment developments," appear redundant and could be rephrased or omitted for brevity.

Response: We are thankful to the reviewer for the advice. In the revised manuscript we have removed the redundant sentences for better clarity.

Comment 2: Mutational Analysis: The study effectively underscores the role of DDX3X in LAML. To enhance the findings, the authors might consider conducting a mutational analysis for DDX3X, akin to the analysis reported for FLT3 and NPM1.

Response: We are thankful to insightful suggestion. In this study we plan to study the role of DDX3X as a target protein for LAML using a comprehensive in silico approach. While we acknowledge the value of performing a detailed mutational analysis—similar to those done for FLT3 and NPM1—our current scope focused on expression profiling, survival correlation, and compound interaction studies to assess DDX3X as a viable therapeutic target. However, we fully agree that mutational characterization of DDX3X, including hotspot identification and co-mutation patterns, would further strengthen the biological relevance of our findings. We intend to incorporate such analyses in future work to complement our current in silico framework.

Comment 3: Plant Name Formatting: Ensure consistent italicization of all plant names throughout the manuscript.

Response: We have italicised the plant name in the revised manuscript.

Comment 4: Rationale for Compound Selection: The authors should elucidate their rationale for selecting natural compounds over synthetic ones for this study.

Response:  We thank the reviewer for the valuable comment. As stated in the introduction, natural compounds from Aerva javanica were selected due to their known anticancer properties, structural diversity, and favorable safety profiles. Given its traditional use and lack of prior studies targeting RNA helicases like DDX3X, A. javanica offered a novel and promising source for identifying potential inhibitors with lower toxicity risks compared to synthetic alternatives.

Comment 5: Justification for Aerva javanica Selection: Please provide a brief explanation in the introduction regarding the selection of Aerva javanica for this study.

Response:  We thank the reviewer for this insightful suggestion. In the revised manuscript, we have added a brief justification for selecting Aerva javanica has now been added to the Introduction. This plant was chosen due to its traditional medicinal use and reported anticancer, anti-inflammatory, and hepatoprotective properties, along with its rich phytochemical profile, which has not been previously explored for DDX3X inhibition.

Comment 6: Exclusion of B2M: Figure 1 indicates that B2M is the most frequently mutated gene in LAML. Please clarify why this gene was excluded from the current study's scope.

Response:  As described in the introduction section, that B2M is the highest mutated gene and well studied as a target protein for various cancer; however, the role of DDX3X- the second highest mutated gene in LAML, was under explored. Therefore, DDX3X was selected for this study to explore into its potential as a target protein for LAML to be inhibited by natural compounds.

Comment 7: Figure 3 Resolution: A high-resolution image for Figure 3 is necessary for publication.

Response:  Thank you for the advice. We have revised the figure.

Comment 8: Tool URLs: Please provide URLs for all computational tools employed in the study.

Response:  Thank you for the advice. We have update the URLs of all the tools used in the study.

Comment 9: In Vitro Validation: The authors may consider conducting in vitro validation studies to further substantiate their findings

Response:  Thank you for the suggestion. The current study is an in silico study and highlighting the importance of in silico studies in the research. Building on these findings, we plan to undertake in vitro validation in future studies to experimentally confirm the predicted interactions, cytotoxicity, and inhibitory effects of the selected phytocompounds on DDX3X function in LAML cell models.